# Actual Measurement and Evaluation of the Balance between Electricity Supply and Demand in Waste-Treatment Facilities and Development of Adjustment Methods

**Daiki Yoshidome** [1], **Ryo Kikuchi** [2], **Yuki Okanoya** [1], **Andante Hadi Pandyaswargo** [3] and **Hiroshi Onoda** [1,*]

[1] Graduate School of Environment and Energy Engineering, Waseda University, 509, 513 Wasedatsurumakicho, Shinjuku-ku, Tokyo 162-0041, Japan; ysdm.3435@akane.waseda.jp (D.Y.); yuuki-0808@fuji.waseda.jp (Y.O.)

[2] Daiei Environmental Research Institute Co., Ltd., 1 Chome-4-1 Nihonbashi, Chuo City, Tokyo 103-0022, Japan; kikuchi9607@dinsgr.co.jp

[3] Environmental Research Institute, Waseda University, 509, 513, Wasedatsurumakicho, Shinjuku-ku, Tokyo 162-0042, Japan; andante.hadi@aoni.waseda.jp

\* Correspondence: onoda@waseda.jp; Tel.: +81-3-6457-3972

**Abstract:** In Japan, breakthroughs to improve the share of renewable energy in the energy mix have become an urgent issue. However, the problem could not be solved by simply adding more power plants for various technical reasons, such as the unsuitability of using renewable energy as baseloads due to its intermittency. Furthermore, establishing the required cooperative systems for regionally distributed power adjustment is also tricky. Based on these backgrounds, this paper constructs an operation plan that minimizes $CO_2$ emissions by correcting the generation and load patterns of the renewable energy of solar power, utilizing power generation from waste as a substitute for baseload power, and estimating the power demand of each facility. The result shows that by adjusting the operation plans, the model can reduce $CO_2$ emission by 20.95 and 8.30% in weeks with high and low solar power generation surpluses, respectively. Furthermore, these results show that it is possible to reduce $CO_2$ emissions in regions that have power sources with low $CO_2$ emission coefficients by forecasting the amount of power generation and power load in the region and appropriately planning the operation in advance.

**Keywords:** waste power generation; energy mix; $CO_2$ emission

## 1. Introduction

In recent years, climate change has become a serious global problem. In Japan, the effects of the Great East Japan Earthquake have spurred interest in renewable energy, and the effective use of energy is increasing [1,2]. Figure 1 depicts the changes in Japan's power source composition before and after the earthquake [3,4]. Before the earthquake, LNG-fired power plants and coal accounted for 29 and 27.8% of the total power supply, respectively, but after the earthquake, the ratio increased to 38.3 and 31.6%, respectively, due to the shutdown of nuclear power plants. In addition, the share of renewable energy increased significantly from 2.2 to 9.2%. In particular, solar and wind power generation is becoming more widespread due to the Feed-in-Tariff (FIT) system [5,6]. However, since these sources are unstable and easily affected by the weather and time, it is not easy to utilize them as a substitute for baseload power supply, such as thermal power generation, which is stable, and its output can be adjusted. In addition, since the FIT system imposes a levy on electricity consumers, the expansion of power sources using this system is limited. It is necessary to consume electricity during power generation rather than by selling electricity by utilizing FIT [7]. Therefore, in recent years, regional utilization of waste power generation has been anticipated. Waste is a stable power source with relatively low $CO_2$ emissions. However, enabling cost-effective waste power generation is challenging due to difficulty securing a waste quantity suitable for incineration, low

power generation efficiency due to a low combustion temperature, and difficulty in power adjustment. In addition, decarbonization in the supply chain under the greenhouse gas (GHG) protocol has been hindered by the increased $CO_2$ emissions due to intermediate treatment facilities, such as incinerators [8,9]. Furthermore, when electricity is purchased from a grid power company in Japan, it is calculated using a constant annual value factor of $CO_2$ emission. However, the $CO_2$ emission factor is not constant because the power supply composition fluctuates depending on the weather and time as the number of unstable power supplies increases [10]. In Japan, the increasing number of unstable power supplies necessitates adjusting the operation of solar power generation during the daytime on holidays. Moreover, the utilization of unstable power supplies is challenging [11–13].

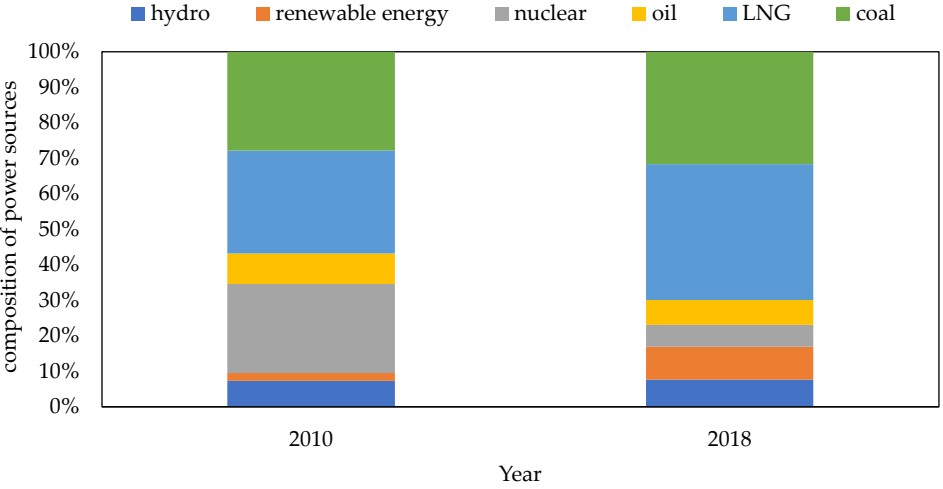

**Figure 1.** Changes in Japan's Energy Mix [3,4].

In this study, to establish a plan to minimize $CO_2$ emissions, large-scale waste treatment/disposal plants (hereafter abbreviated as RC) located in Iga City, Mie Prefecture, and solar power generation equipment (hereafter abbreviated as PV) located in Izumi City, Osaka Prefecture and owned by the same group corporation are taken as case study facilities. The RC plants were considered as one block (RC block). The internal processing equipment represented the consumer, and the three incineration facilities and PV in the RC block were the power supply source.

Here, the present study established a method to predict power load and power generation patterns by acquiring and analyzing power demand and supply data at each facility. In Japan, methods for predicting energy demand in specific city blocks and the amount of electricity generated by specific facilities have been established [14,15]. However, these methods are designed to be used in business models based on the viewpoint of power generators and retailers who supply electricity, and few studies capture the matching of supply and demand from both directions. One approach similar to this study is peak shifting, but the peak shifting plans that have been popularized mainly by retail electric utilities mostly only change the electricity price depending on the season and do not change the demand-side energy use plan following the demand-side energy use plan with the real-time power supply composition. Similarly, changing the demand-side energy use plan is impossible to comply with the real-time power supply configuration [16]. In a demand response approach, where consumer electricity consumption is reduced by fluctuating electricity prices when electricity load is tight, using simulation models in residential areas is effective. However, only a few industrial companies have taken this approach, and its actual status and effectiveness are unclear [17,18]. The novelty of this research lies in the fact that, based on this background, this study has designed a model that makes it possible to construct electricity procurement and usage plans that minimize $CO_2$ emissions based on measured data from actual operating facilities in the industrial sector and the demand and generation characteristics of each facility. In addition, another method was used to

calculate a CO$_2$ emission factor that fluctuated with time (hereafter termed CO$_2$ emission factor by time of the day), considering that the consumption rate of renewable energy in the domestic electricity market fluctuates with time. Finally, this study predicted weekly CO$_2$ emissions using the load pattern of power supply and demand and the time-of-day CO$_2$ emission factor, created an operation plan with the smallest CO$_2$ emissions and verified its effects. In addition, while other countries can procure electricity from land-linked areas, in reality, all countries still face the issues of what to do with nuclear power and how to stably supply electricity from renewable energy sources [19]. In Japan, such issues became apparent first in the world in the wake of the Great East Japan Earthquake [20]. Therefore, this study is presented to serve as a model both for the power adjustment in Japan and for other countries by providing a clue to the power supply problems that many countries will face in the future.

## 2. Materials and Methods

Waste in Japan is roughly divided into two types: industrial waste generated by business activities and general waste [21]. Industrial waste is classified into 20 items according to its nature, and the processing method differs depending on the item. The RC comprises many treatment facilities that enable the treatment of many industrial and general wastes and consists of a controlled final disposal site, making it the largest waste treatment and disposal facility in Japan [22,23]. Because there are many types and incoming quantities of waste, the operation plan of each processing facility is alternated based on the weekly waste delivery schedule. Figure 2 depicts the material flow of RC waste. Since RC also accepts disaster-related waste, the waste load varies depending on the time of year.

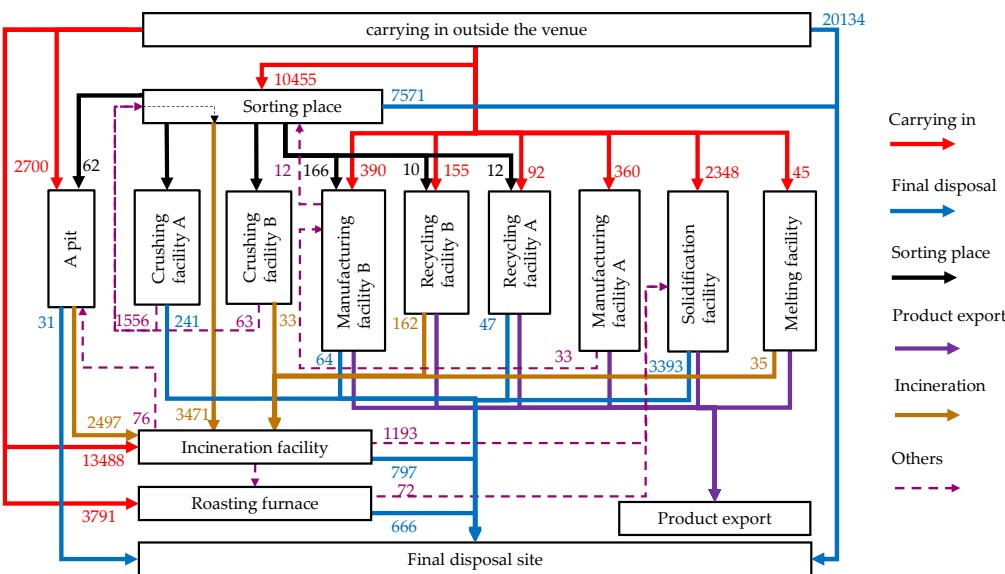

**Figure 2.** Material flow in RC.

Nonetheless, except on Sundays, approximately 2000 t of waste is delivered daily. Table 1 shows the items handled and the processing capacity of each facility. Figure 3 shows the relationship between energy supply facilities and energy demand facilities in the RC. Incineration facility A in RC ("Incineration facility A") generates steam turbine power using the heat from the incinerated waste ("Steam turbine α"). In addition to the steam turbine power generation that uses the excess exhaust heat of incineration facility A ("Steam turbine β"), incineration facility B ("Incineration facility B") also uses steam turbines to generate electricity. The electricity generated by the three facilities is used as the operating power of each processing facility in the RC, and surplus electricity is sold to the grid power company. Figure 4 depicts the energy balance of Steam turbine α. The amount of heat generated from the boiler is 27,740 MJ/h, of which 38% is used to power the turbine.

In addition to power generation, heat is used for wastewater concentration in the water treatment facility, a trans-heat container that stores heat, generating approximately 10% excess heat [24].

**Table 1.** Items handled and processing capacity of each line.

| Processing Facility | Items Handled | Processing Capacity, t/D |
|---|---|---|
| Crushing facility, A | Bulky waste | 250 |
| Crushing facility B | Bulky waste | 98.4 |
| Recycling facility A | Small home appliances | 30 |
| Recycling facility B | Plastic packaging and containers | 25 |
| Manufacturing facility A | Wood waste | 262 |
| Manufacturing facility B | Plastic and paper waste | 69 (×2) |
| Solidification facility | Soil and contaminated soil | 400 |

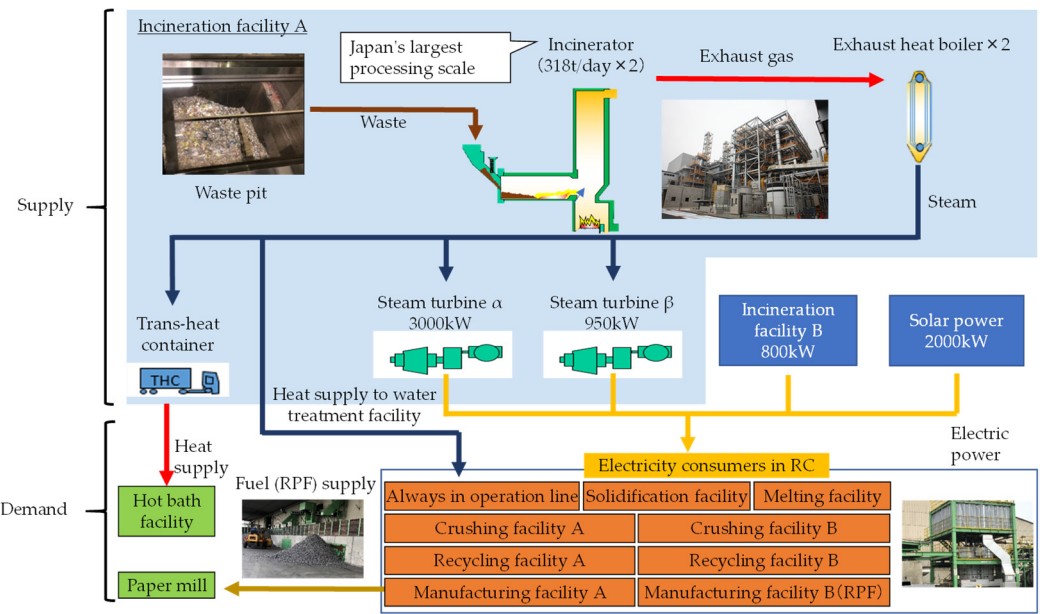

**Figure 3.** RC energy flow.

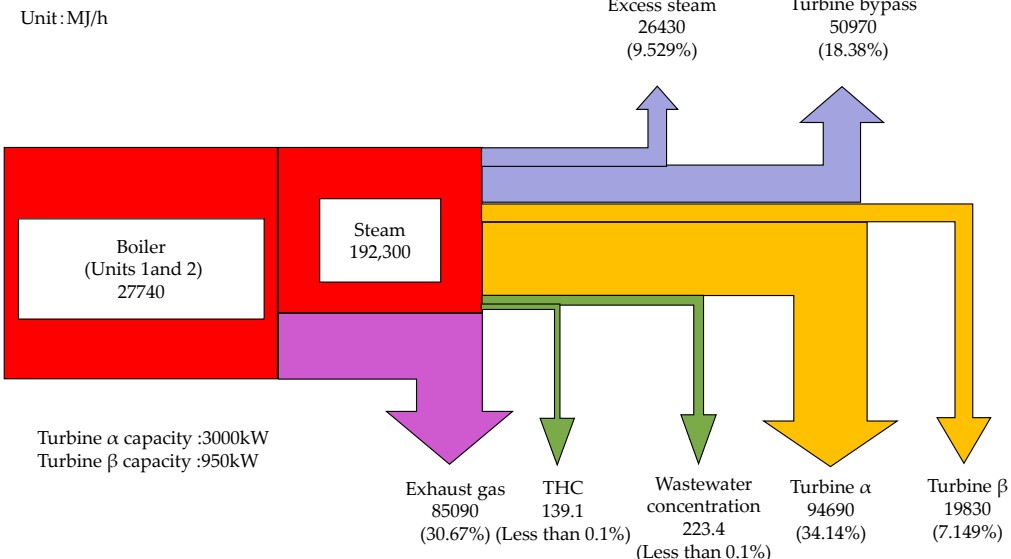

**Figure 4.** Energy balance.

This research is conducted in three stages: (1) forecasting power generation and power load patterns, (2) forecasting $CO_2$ emission factors, and (3) constructing an operation plan that minimizes $CO_2$ emissions.

The target of this forecast for the power generation facilities can be broadly classified into two main categories: solar power generation and waste incineration power generation. The waste incineration power generation can be predicted from the steam generation of the boiler and the power generation efficiency. However, since most incineration plants keep the calorie content of the waste incinerated at a constant level to stabilize the combustion conditions, it is assumed that the power generated will also be stable [25]. Therefore, this study is aimed to correct the power generation pattern by obtaining the amount of power generated by waste incineration every minute. In general, there are two methods for predicting PV power generation: one is to predict the amount of power generation directly from weather conditions, and the other is to predict the amount of power generation indirectly from solar radiation [26]. Therefore, this study analyzed the power generation pattern by measuring the weather conditions, the amount of power generated per minute, and solar radiation in the PV area. Furthermore, it is assumed that the power load is directly affected by the operating hours of the equipment. Here, the study analyzed the power load pattern for each facility by obtaining the daily operating hours and the power load per 10 s for each facility.

In addition, a field survey was conducted because it is expected that the power load and power generation amount during the operation of each facility will have characteristic features. The results of the field survey are shown in Table 2. The field survey revealed that RC is roughly divided into facilities that are always operational regardless of the amount of waste carried in (henceforth, "always in operation line") and facilities that create an operation plan according to the amount of waste carried in (henceforth, "operation plan line"). Based on the survey results, the operation plan of the facility will be developed to minimize $CO_2$ emissions by adjusting the operation time of the planned operation line.

Next, the $CO_2$ emission factor by time of the day was calculated. As mentioned above, a calculation method used in Japan involves using a constant factor in calculating $CO_2$ emissions from electricity throughout the year [27]. However, the actual $CO_2$ emission factor differs between daytime (when the ratio of solar power generation is large) and nighttime (when the ratio of thermal power generation is large). Furthermore, since the demand for electricity and the amount of solar power generation fluctuate during the year, the $CO_2$ emission factor fluctuates according to the time of day. Therefore, in this paper, the $CO_2$ emission factor with time was calculated from the power generation results in the past five years of the grid power company contracted by RC (henceforth, "power company A"). Furthermore, the fluctuation factors of $CO_2$ emission factor by time of the day were analyzed, and a prediction method was developed.

Finally, to minimize RC $CO_2$ emissions, an operation plan was created using optimization calculations from the load pattern, power generation pattern, and $CO_2$ emission factor by time of the day. There are five main elements in the operation standard of the operation plan line: residual storage capacity of the unprocessed stockyard to store unprocessed waste (henceforth, "B-SY"), the residual storage capacity of the post-processing stockyard to store waste after processing (henceforth, "A-SY"), scheduled amount of incoming/outgoing waste, securing vehicles for unloading, and securing workers. In the present study an operation plan that minimizes $CO_2$ emissions by considering the three factors of B-SY, A-SY, and planned loading/unloading amounts for the operation plan line was created. However, it is difficult to obtain data because each facility's operation plan is made separately. Therefore, the $CO_2$ emission reduction effect was calculated by comparing the $CO_2$ emissions estimated from the operating time based on the operating results and those after optimization. Figure 5 presents the operation plan that minimizes $CO_2$ emissions.

**Table 2.** Priorities in the treatment process.

| Processing Facility | Priority in the Processing Process | | | | Remarks |
|---|---|---|---|---|---|
| | Carry-in Schedule | B-SY | A-SY | Carry-Out Schedule | |
| Crushing facility A | × | × | × | ○ | · Adjusted to meet the fuel demand of the power plant<br>· Can be operated for half a day |
| Manufacturing facility A | ○ | △ | × | ○ | · Consider the destination schedule<br>·Acceptance is required |
| Recycling facility A | ○ | ○ | × | × | · The capacity is only 10 t, but it cannot operate unless it is filled fully<br>·No storage capacity on the carry-out side |
| Manufacturing facility B | ○ | △ | × | × | · Even if the amount of delivery is not enough, it will be processed as soon as feedstock is received<br>· It is necessary to take measures at the time of acceptance |
| Solidification facility | △ | △ | △ | × | ·No effect on operation even if the facility is stopped temporarily |
| Recycling facility B | ○ | × | × | × | · Annual delivery schedule can be recognized here<br>· Feedstocks are processed immediately |
| Crushing facility B | × | × | × | × | · Must be operated when a person is present |

"X" indicates no consideration, "△" indicates consideration is necessary in some cases, and "○" indicates consideration is necessary.

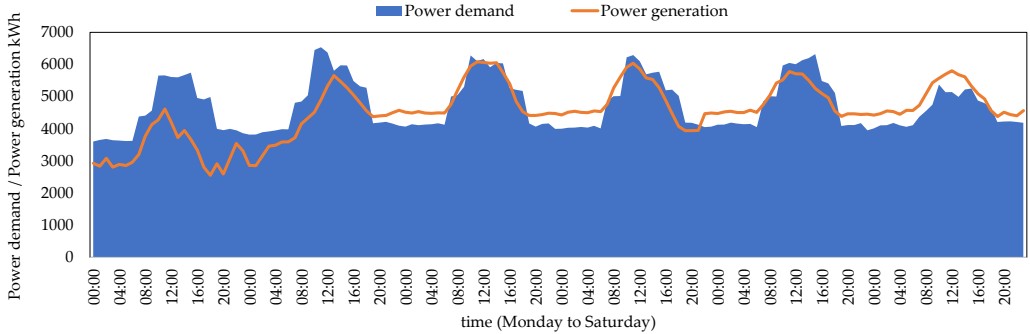

**Figure 5.** Image of operation shift.

## 3. Research Results

### 3.1. Analysis of Power Supply and Demand at Each Facility

As mentioned above, the load pattern can be divided into operation-plan line and always-in-operation line, and the power generation pattern can be divided into waste and solar power generation. Figures 6–9 show the change in power demand during regular operation on a specific day. The operation plan line clearly shows the power load during non-operation and operation. On the other hand, the always in-operation line requires a constant power load throughout the day. In addition, waste power generation is stable throughout the day, while the amount of power generated by solar power generation fluctuates greatly depending on the weather and time of day. Therefore, in this paper, the load pattern and power generation pattern are roughly divided into operation plan line, always in operation line, waste power generation, and solar power generation.

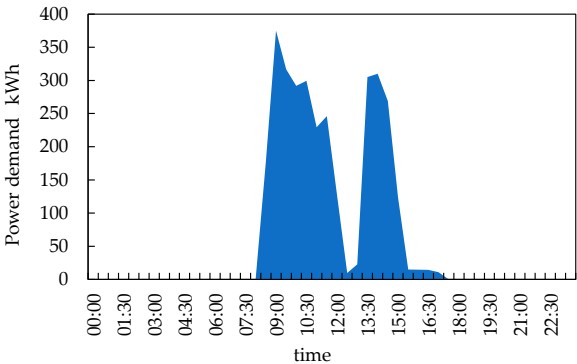

**Figure 6.** Operation planning line (crushing facility A) load pattern.

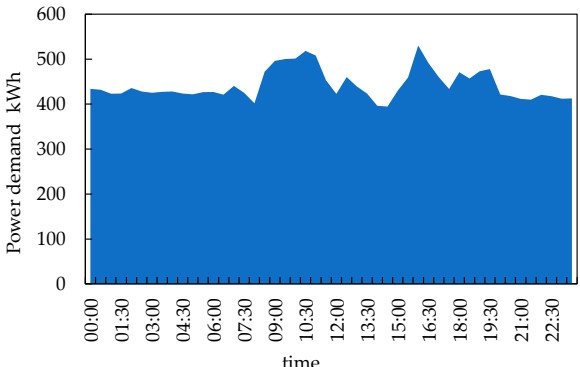

**Figure 7.** Load pattern of continuous operation line (a new water treatment facility).

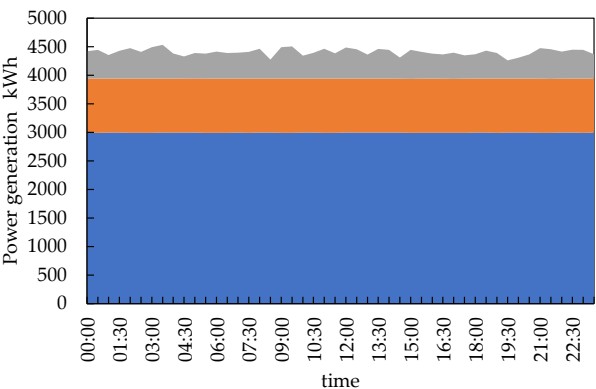

**Figure 8.** Amount of power generated by waste power generation facilities.

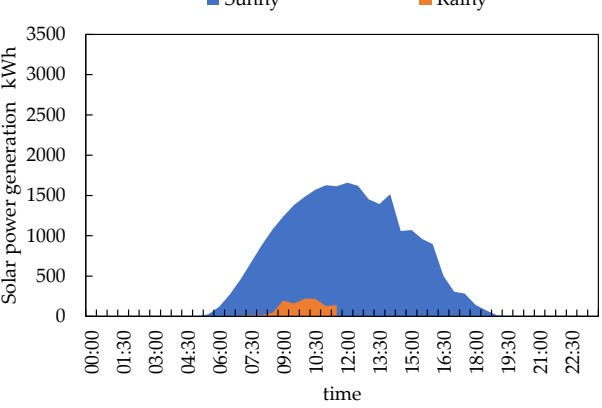

**Figure 9.** Amount of solar power generation by weather.

### 3.1.1. Development of Load Pattern for Operation Plan Line

As shown in Figure 10, when the power load of the operation plan line was analyzed, it was expressed by the step values during non-operation and operation. It was evident that the power load during operation differed depending on the processing method of the facility. Therefore, Equation (1) calculates the daily load pattern. In addition, the operating power, non-operating power, and the prediction errors of each facility are shown in Table 3.

$$P_c = P_1 \times A_{1(t)} + P_2 \times A_{2(t)} + \cdots + P_n \times A_{n(t)} \tag{1}$$

where, $P_c$ is the consumed power (kW), $P_{1 \sim n}$ is the power value for conditions 1 through $n$ (kW), $A_{1(t)}$ is the binary value for conditions 1 through $n$ at $t$ 0 or 1, and $t$ is time.

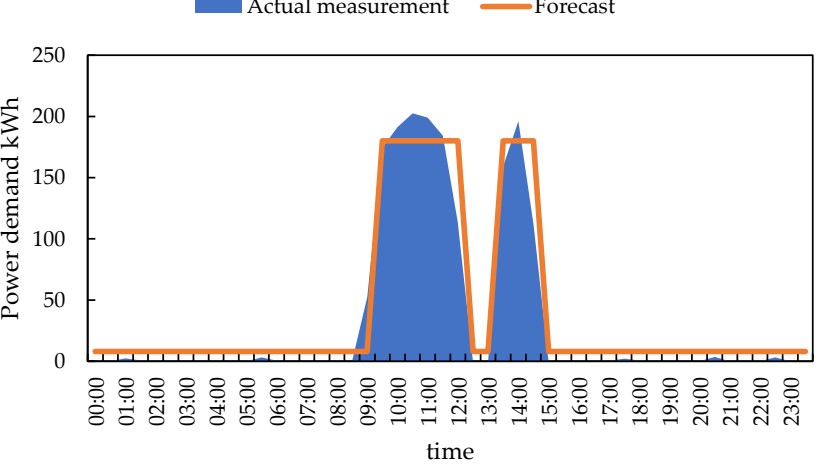

**Figure 10.** Operation planning line (Manufacturing facility A) load pattern.

**Table 3.** Operating power and error rate of each facility.

| Line Name | Operation Power, kW | Non-Operation Power, kW | Prediction Error, % |
|---|---|---|---|
| Crushing facility, A | 265 | 0 | 13.68 |
| Manufacturing facility B | 746 | 2 | 9.13 |
| Manufacturing facility A | 158 | 8 | 12.41 |
| Home appliances | 230 | 6 | 19.33 |
| Solidification facility | 162 | 0 | 16.84 |
| Container packaging | 379 | 35 | 11.66 |
| Crushing facility B | 73.5 | 0 | 18.27 |

### 3.1.2. Development of Load Pattern of Always-in-Operation Line

As shown in Figure 11, in the always-in-operation line, the operating power load fluctuates even during 24 h operation. A field survey of this factor revealed that the equipment of analysis of the power load during operation of each always in operation line are shown in Table 4.

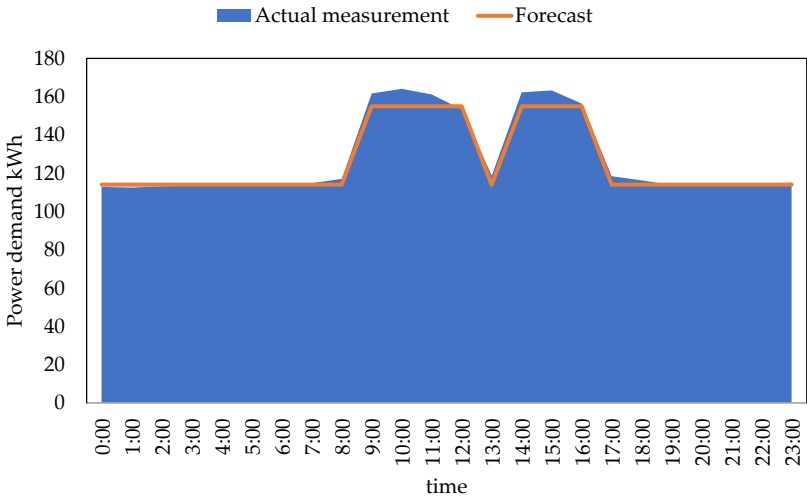

**Figure 11.** Load pattern of continuous operation line (Roasting pretreatment facility).

**Table 4.** Power demand and error rate of always-online.

| Line Name | Basic Power, kW | Nighttime Power, kW | Holiday Power, kW | Prediction Error, % |
|---|---|---|---|---|
| Roaster | 449 | 411 | 411 | 5.16 |
| Roaster pretreatment | 155 | 114 | - | 12.61 |
| Water treatment | 128 | 100 | - | 11.33 |
| New water treatment | 350 | 270 | - | 10.10 |
| Office | 105 | 53 | 53 | 13.35 |
| Incinerator B (operation power) | 885 | - | - | 1.84 |
| Incinerator A (operation power) | 2190 | 2040 | 2040 | 3.08 |

### 3.1.3. Development of Power Generation Pattern for Waste Power Generation

Regarding waste power generation, as mentioned above, three facilities are operating in RC. Steam turbine $\alpha$ and Steam turbine $\beta$ have a constant amount of power generation because the amount of steam is always in surplus, and power is generated up to the upper limit, as shown in Figure 4. Equations (2) calculate the amount of power generated by Steam turbine $\alpha$ and Steam turbine $\beta$.

$$P_s = \overline{P_s} \tag{2}$$

where, $P_s$ is the waste power generation in kW, and $\overline{P_s}$ is the constant power generation value.

Each incineration facility periodically undergoes repair work, and a one-sided furnace operation is carried out for dozens of days throughout the year. Since the amount of steam generated decreases during single-fired operation, the amount of power generated by Steam turbine $\alpha$ and Steam turbine $\beta$ differs from that during regular operation. The amount of power generated during single-fired operation is shown in Table 5.

**Table 5.** Power generation of steam turbines α and β.

| Facility | Power Generation, kW | Power Generation during Single Furnace Operation, kW |
|---|---|---|
| Steam turbine α | 2993 | 2690 |
| Steam turbine β | 944 | 0 |

In incineration facility B, the amount of power generated fluctuates due to the variable amount of steam generated. In addition, the soot blower operates by utilizing the generated steam to remove soot and dust, adhering to the electric heating surface of the boiler. Therefore, as shown in Figure 12, the amount of power generated during operation fluctuates regularly. The amount of power generated by incineration facility B is shown in Table 6.

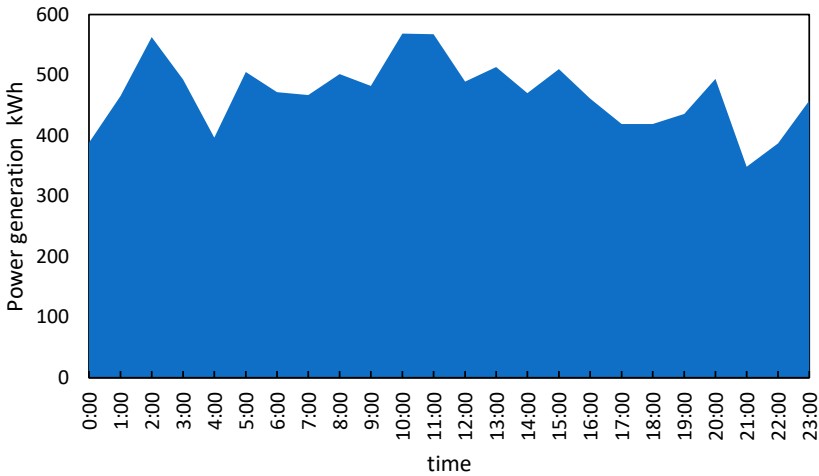

**Figure 12.** Incineration facility B.

**Table 6.** Power generation of incinerator B.

| Facility | Power Generation, kW | During Soot Blower Operation, kW |
|---|---|---|
| Incineration facility B | 530 | 420 |

### 3.1.4. Examination of the Prediction Method for Solar Power Generation Amount

The amount of solar power generation depends on the fluctuation of solar radiation, which depends on the weather conditions and the time of day. Therefore, in this paper, a method to estimate the amount of solar radiation to predict the amount of power generated in the week when the operation plan was developed. Multiple operators provide services aimed at forecasting the amount of solar radiation. Therefore, this study examined the Japan Meteorological Agency's solar radiation forecast API (Application Programming Interface) and weather forecast, since private businesses can freely use it. In Prediction method 1, the prediction from API was used to predict the amount of solar radiation. On the other hand, in Prediction method 2 prediction from the weather forecast was used. The error that occurred when the prediction was made weekly was evaluated. Solar radiation amount prediction API can predict the amount of solar radiation up to 3 days later in 1 h units and the amount of solar radiation up to 7 days later in 3 h units. Therefore, in Prediction method 1, the weekly solar radiation prediction result for 1 h was calculated by complementing the solar radiation prediction result for 3 h. Additionally, in Prediction method 2, the weather forecast conducted one week in advance, from 11 February to 11 March, 2020, was classified as sunny, cloudy, rainy, or sometimes cloudy (Table 7) and then the average value for each category was calculated. Using the results, each solar

radiation pattern was determined. Figure 13 shows an example of the comparison of the measured values of Prediction methods 1 and 2. The average error of the power generation amount calculated by Prediction methods 1 and 2 during the daytime (9:00 to 18:00) from 11 February to 11 March 2020 was 116.0 and 125.3 $W/m^2$, respectively. Both values accounted for approximately 3% of the total amount. Based on the weather forecast information results, this study used Prediction method 2, which is the easiest to use. However, since the solar radiation pattern varies depending on the season, the solar radiation pattern for each month was created to calculate the amount of power generation. The amount of solar radiation and power generation displayed a positive correlation. The process of calculating the amount of power generation from the amount of solar radiation is described in Equation (3):

$$P_{solar} = B \times S_r \tag{3}$$

where, $P_{solar}$ is the solar power generation [kW], B is the power generation factor (=1.91) [$kW/(W/m^2)$], and $S_r$ is the predicted amount of solar radiation [$W/m^2$].

**Table 7.** Weather classification.

| Weather Forecast | Sunny | Cloudy | Rainy | Cloudy but Occasionally Sunny |
|---|---|---|---|---|
| Actual weather | Sunny<br>Clear at one point<br>Sunny in the morning<br>Sometimes sunny | A little cloudy<br>A little cloudy at one point<br>A little cloudy in the morning<br>Sometimes a little cloudy<br>Cloudy<br>Cloudy at one point<br>Cloudy in the morning | Rainy<br>Rainy at one point<br>Rainy in the morning<br>Sometimes raining<br>Heavy rain<br>Cloudy sometimes raining | Cloudy butoccasionally sunny |

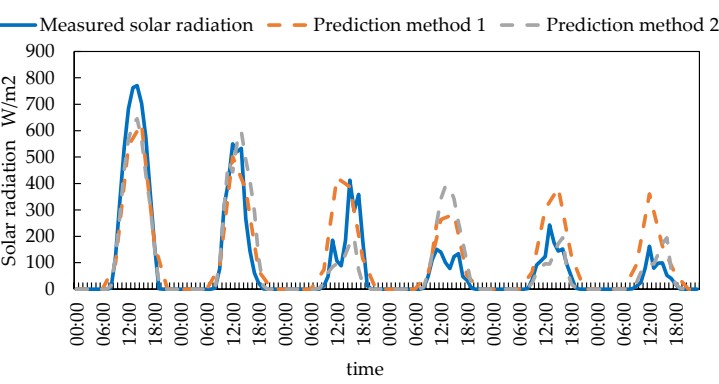

**Figure 13.** Comparison of solar radiation prediction methods.

*3.2. Development of Prediction Method for $CO_2$ Emission Factor by Time of the Day*

In calculating the $CO_2$ emission factor by time of the day, this study analyzed the supply and demand performance of power company A (April 2016 to September 2020). Figure 14 shows the actual power supply on 1 April 2020, as an example of the breakdown of the power supply. Power company A is one of the largest power generation companies in Japan and owns multiple power sources. Therefore, in calculating the $CO_2$ emission factor by time of the day, the $CO_2$ emission factor at the transmission end of each power source is shown in Table 8. The emission factor of thermal power generation was calculated based on the composition ratio of coal and petroleum liquid natural gas (LNG). In addition, core emission factors and adjusted emission factors are relevant when calculating the $CO_2$ emission factor. However, since this section does not evaluate the environmental value of power company A, this study uses the concept of core emission factors. The calculation method of the core emission factor is shown in Equation (4). From the actual data on area

supply and demand, the $CO_2$ emission factor in power company A for each time point was calculated using Equation (5).

$$CEF = \frac{Q}{E} \qquad (4)$$

where $CEF$ is the core emission factor (t-$CO_2$/kWh), $Q$ is the core $CO_2$ emission amount (t-$CO_2$), and $E$ is the electric power sold (kWh).

$$p = \frac{\sum t_i \cdot u_i}{k} \qquad (5)$$

where $p$ is the $CO_2$ emission factor by time of the day (kg-$CO_2$/kWh), $t_i$ is the amount of power generated by each power source (MWh), $u_i$ is the $CO_2$ emission factor of each power source (kg-$CO_2$/kWh), and $k$ is the area demand (MWh).

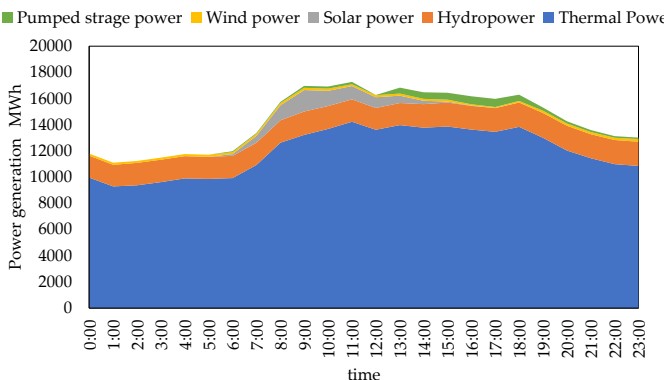

**Figure 14.** Power generation record of electric power company A.

**Table 8.** The $CO_2$ emission factor of each power source.

| Power Plant | $CO_2$ Emission Factor, kg-$CO_2$/kWh |
|---|---|
| Thermal power plant | 0.4997 |
| Solar power | 0.038 |
| Wind farm | 0.026 |
| Hydroelectric power plant | 0.011 |
| Adjustments (e.g., FIT) | 0.490 |
| Pumped-storage hydropower | 0.011 |

It was found that only the solar power generation fluctuated significantly in the power supply composition by season and time of the day. Based on this finding, multiple regression analysis was performed to correct the correlation between solar power generation and $CO_2$ emission factor by time of the day for power company A. As shown in Figure 15, there was a high correlation ($R^2$ = 0.8115) between the coefficient of determination of solar power generation amount and $CO_2$ emission factor with time of the day. Therefore, the present study decided to predict the $CO_2$ emission factor by time of the day after predicting the solar power generation amount of power company A using the weather forecast of Nagoya, which is the power generation area of power company A.

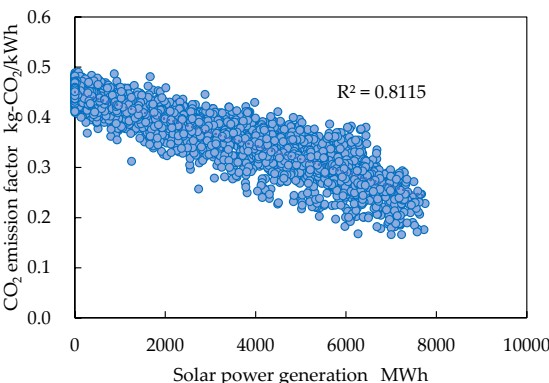

**Figure 15.** Relationship between solar power generation and $CO_2$ emission factor.

### 3.3. $CO_2$ Reduction Effects due to Operation Shift

To analyze the elements related to the current operation plan, a multiple regression analysis was performed on the amount of waste carried in and the weekly operation plan for the operation plan line data. The analysis revealed a positive linear correlation (Figures 16 and 17). This result agrees with the results of the field survey. Therefore, it was evident that the import quantity is one of the important elements of the operation plan. In addition, a similar multiple regression analysis revealed a positive linear correlation between the operating time and electric energy (Figures 18 and 19). These collective findings indicate that the operating time can be calculated from the weekly carry-in amount, and the weekly electric energy can be predicted from the operating time.

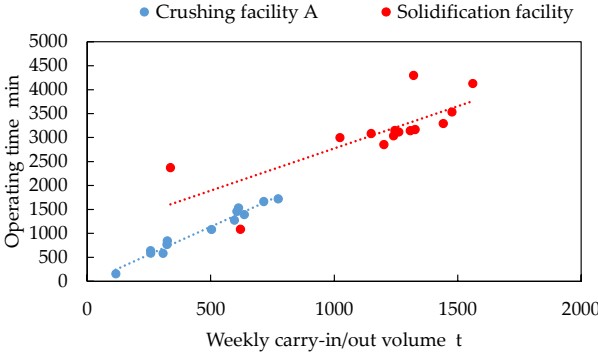

**Figure 16.** Relationship between the amount of crushing facility A and solidification facility input and operating time.

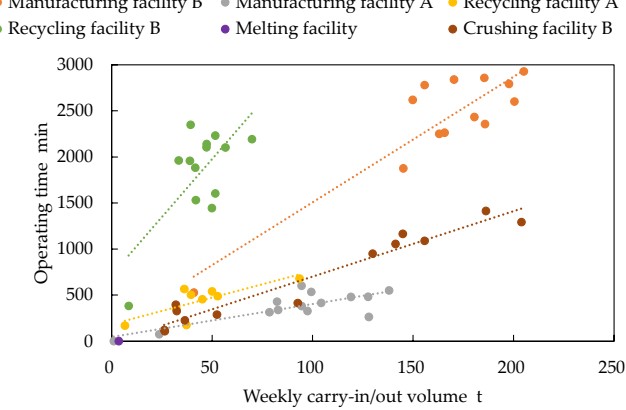

**Figure 17.** Relationship between the carry-in/out volume and operating time of facilities other than crushing facility A and solidification facility.

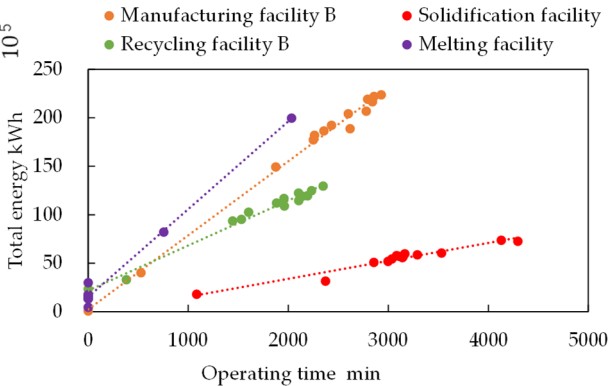

**Figure 18.** Relationship between operating time and weekly operating power.

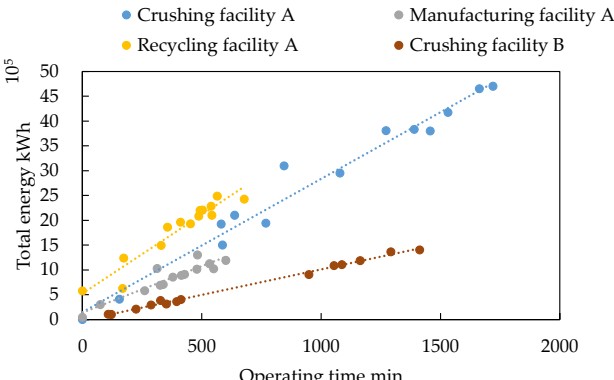

**Figure 19.** Relationship between operating time and weekly operating power.

Minimizing the amount of $CO_2$ reduction by altering the operation plan of the operation plan line was explored by mathematical optimization. The model equation for optimizing calculation is shown in Equation (6). From Equation (6), the operation judgment of each processing facility is binary, so it is non-linear. Therefore, the calculation was performed using the generalized reduction gradient method and the genetic algorithm. The constraints are shown in the following Table 9. The operation plan line comprises seven more facilities than the always in operation line (Roaster, Roaster pretreatment, Water treatment, New water treatment, office, Incinerator B (operation power), Incinerator A (operation power)), and the operating power uses the setting values obtained in Table 4. In addition, the actual measurement value is used for the operating time of each processing facility.

$$EM_{(x_i, \ CEF_t)} = \left( \sum_{i=1}^{8} A_i \cdot x_i - EP \right) \times CEF_t \qquad (6)$$

where $EM$ is the $CO_2$ emission amount (kg-$CO_2$/W), $A_i$ is the operating power of each treatment facility (kWh/W), $x_i$ is the operation decision of each treatment facility (0 or 1), and $EP$ is the power generation amount (kWh/W).

Figure 20 illustrates how to create an operational plan that minimizes $CO_2$ emissions. In this study, actual data are used in some calculation processes because the verification does not include understanding operating hours and operating schedules. Figure 21 shows the results of the operation shift from 13 April (Monday) to 18 April (Saturday). Before the shift to operation, the amount of power generated on Monday was small, which was below the power demand of the processing equipment. The difference between power generation and power demand was relatively low from Tuesday to Friday. Approximately 500 kW of surplus electricity was generated on Saturday. When the operation shift was performed, the operation on Monday turned to Saturday, and the power usage from the

grid on Monday was significantly reduced. The amount of reduction was 2520 kg-$CO_2$ before and after the shift, and the reduction rate was 20.95%.

**Table 9.** Constraints on the operation of each facility.

| Processing Facility | Constraints | Processing Plan Criteria |
|---|---|---|
| Crushing facility A | — | |
| Manufacturing facility B | Securing the amount to be carried out | |
| Manufacturing facility A | Securing the amount to be carried out | Achievement of the weekly processing target |
| Recycling facility A | Securing SY before processing | |
| Solidification facility | — | |
| Recycling facility B | — | |
| Crushing facility B | — | |

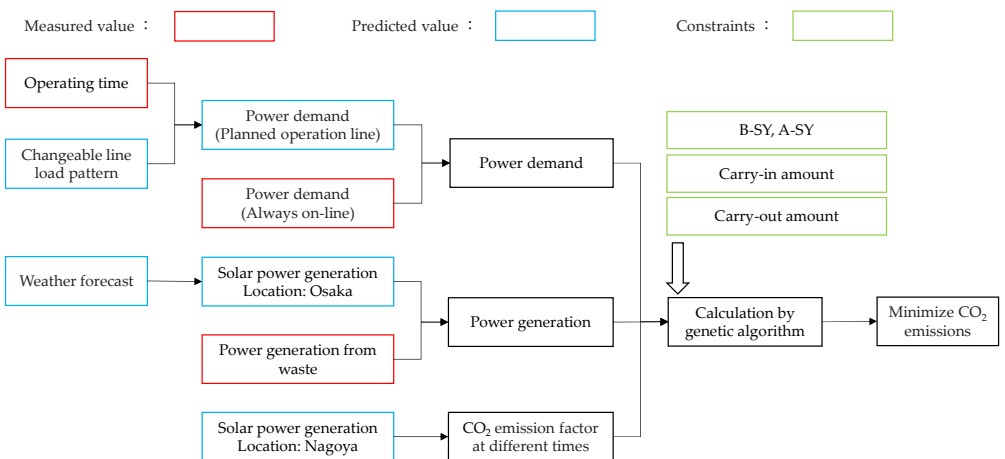

**Figure 20.** The logic for calculating planned operating values.

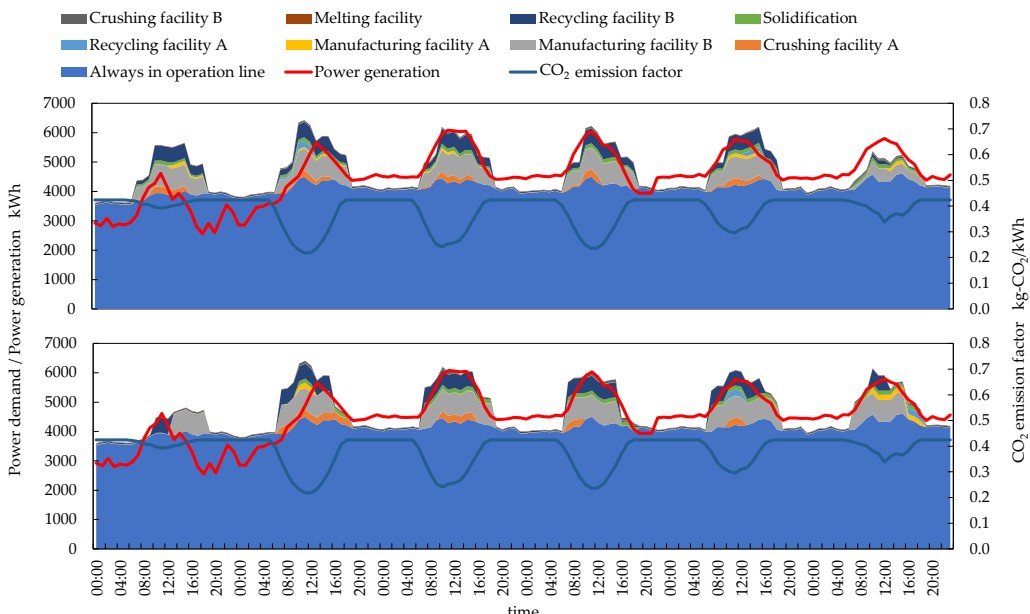

**Figure 21.** Results of weekly operational shifts where surplus power is generated.

Figure 22 shows the results of the operation shift from 15 to 20 June. The weather was cloudy throughout that week, and the fluctuation in the amount of solar power generation

was small. As a result of the operation shift, operations were concentrated on Saturday, and the reduction rate was 8.30%.

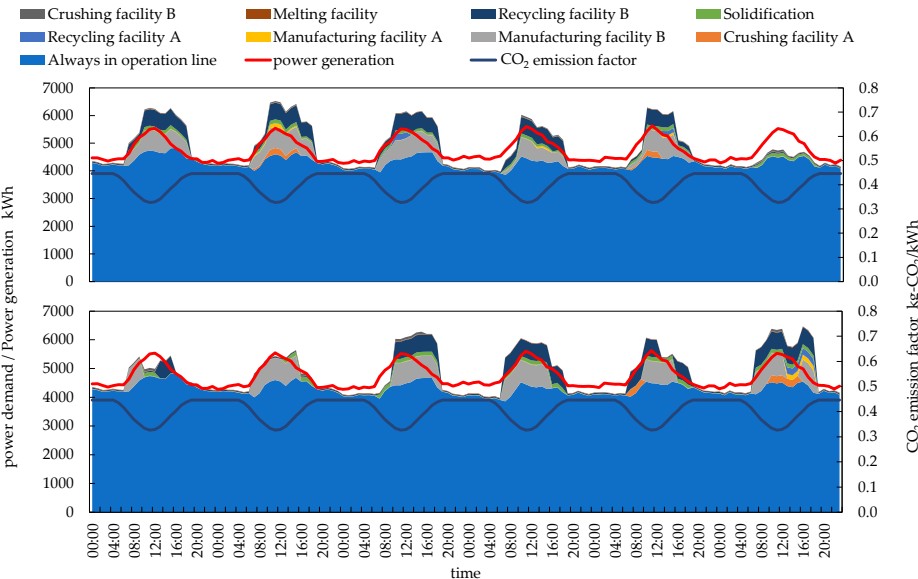

**Figure 22.** Weekly operation shift results with no surplus power.

## 4. Discussion

This study aimed to construct a model for operation planning that minimizes $CO_2$ emissions. The research results show that it is possible to construct an operation plan that minimizes $CO_2$ emissions by predicting the amount of electricity generated, the amount of demand, and $CO_2$ emissions. Although the subject of this study is the power load of photovoltaic power generation, waste power generation, and waste treatment facilities, the present study believes that it is highly likely that power usage planning that minimizes $CO_2$ emissions can be established for other facilities as well, if sufficient prediction accuracy can be obtained. On the other hand, this study found that in some cases, the forecasts of solar power generation and $CO_2$ emission factor by time of the day may not be effective enough because the accuracy of the weather forecast affects the error in $CO_2$ emissions. To reduce this error, this study examined the improvement in the prediction accuracy of solar radiation by using data from a field survey to modify the operation plans. Figure 23 shows a comparison of the forecast results and actual measurements using the weather forecast for one week and two days in advance for photovoltaic power generation and the $CO_2$ emission factor. The $CO_2$ emission factor shows a tendency similar to the predicted result, but it can be confirmed that the amount of photovoltaic power generation is different from the actual measurement. Therefore, each absolute error was calculated and shown in Table 10. On the other hand, it was found that there is no significant difference between the prediction results one week and two days in advance with the current prediction method. In future research, reducing the absolute error by further improving the prediction method will be considered.

**Table 10.** Absolute error in actual measurement and prediction of solar power generation and $CO_2$ emission factor.

|  | Solar Power Generation (Absolute Error), kWh | $CO_2$ Emission Factor (Absolute Error), kg-$CO_2$/kWh |
|---|---|---|
| Forecast one week in advance | 156 | 0.018 |
| Forecast two days in advance | 137 | 0.020 |

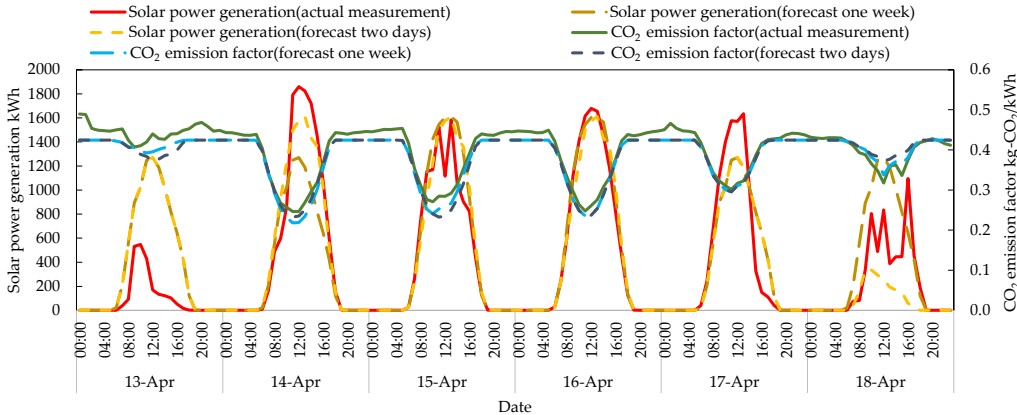

**Figure 23.** Comparison of solar power generation and $CO_2$ emission factor made one week and two days in advance.

## 5. Conclusions

Due to the intermittency and adjustability of popular renewable energy such as solar and wind power, they are unsuitable for baseload power supply substitute. In recent years, Japan has been promoting power generation from municipal waste for regional utilization. Waste is considered a stable power source with relatively low $CO_2$ emissions. However, several factors have hindered cost-effective waste power generation. Some of these factors are: (1) difficulty in securing a waste quantity suitable for incineration, (2) low power generation efficiency, and (3) difficulty in power adjustment. In addition, $CO_2$ emissions are calculated imprecisely because the waste input fluctuation over the years was not considered.

Responding to the above challenges and supporting better waste power generation utilization, this study took a large-scale waste treatment plant in Iga City, Mie Prefecture, and a solar power generation plant located in Izumi City, Osaka Prefecture as case study plants. Using actual data from the case plants, this study has:

- Developed a prediction method by analyzing the power data of operation plan line, always in operation line, waste power generation, and solar power generation. For solar power generation, the average error was 125.3 kW by predicting the amount of solar radiation based on the weather forecast.
- Calculated the $CO_2$ emission factor by time of the day from the data of power company A analysis. This study also developed a method to predict the $CO_2$ emission factor by time of the day with a prediction accuracy of 0.81 by predicting the solar power generation amount in power company A.

Results showed that by formulating an operation plan for the operation plan line using $CO_2$ emission factor by time of the day and a method for predicting power load and power generation, $CO_2$ emissions could be reduced by 20.95% in weeks with high solar power generation and by 8.30% in weeks with low high solar power generation.

The limitation of this study was that the actual operation plan could not be acquired (this study only estimated it using the power data due to company confidentiality). Furthermore, personnel constraints were not considered in the proposed operation plan.

When constructing an operation plan of a similar power generation facility with the purpose of $CO_2$ emissions reduction, it is necessary to accurately predict the following three parameters: (1) power generation, (2) power demand, and (3) $CO_2$ emission factor. Thus, while the prediction method developed in this study can be used as a benchmark, future works analyzing other facilities must employ the actual parameters from the specific plants. Additionally, the target power load facilities in this study are limited to industrial facilities in the industrial sector. Actual users in the region include residential and commercial facilities, hospitals, and public facilities. The findings of this study can be used to establish

a forecasting method for waste treatment facilities and to plan electricity use that minimizes $CO_2$ emissions as a region.

**Author Contributions:** All the authors have contributed substantially to the work reported. Conceptualization, D.Y. and H.O.; methodology, D.Y., R.K. and Y.O.; formal analysis, D.Y., R.K. and Y.O.; investigation, D.Y., R.K. and Y.O.; data curation, D.Y., R.K. and Y.O.; writing—original draft preparation, D.Y. and H.O.; writing—review and editing, D.Y., R.K., Y.O., A.H.P. and H.O. All authors have read and agreed to the published version of the manuscript.

**Funding:** This research received no external funding.

**Institutional Review Board Statement:** Not applicable.

**Informed Consent Statement:** Not applicable.

**Data Availability Statement:** The data presented in this research are available on request from the corresponding author.

**Conflicts of Interest:** The authors declare no conflict of interest.

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
