# Peer review of "Actual Measurement and Evaluation of the Balance between Electricity Supply and Demand in Waste-Treatment Facilities and Development of Adjustment Methods"

_applsci, doi:10.3390/app112210747_

Round 1

Reviewer 1 Report

  • The Authors need to get editing help from someone with full professional proficiency in English
  • Abstract: “…can reduce CO2 by 20.95% and 8.30% in weeks with high and low 21 PV surpluses, respectively.”

Reduce CO2? What does PV stand for?

  • Section 1 is extremely poorly written. In this Section the Authors need to describe the current status on the investigated topic and then clearly state what knowledge this work will fill COMPARED TO the current status on the investigated topic
  • How and why should this work draw the attention of researchers from outside Japan?
  • This manuscript seems to be a very interesting report rather than a scientific article
  • The results showed in Figure 1 need to be supported by suitable references
  • Passive voice needs to be adopted in the whole manuscript
  • Units of measurement are incorrectly formatted in the whole manuscript (e.g. kg*s^-1 rather than kg/s)
  • The accuracy of the experimental equipment as well as the uncertainty propagation analysis are missing
  • There is no discussion in Section 4
  • 2 of CO2 is a subscript

Author Response

Cover letter

November 5, 2021

Environmental Research Institute

Waseda University, Japan

Dear Applied science Editor,

We would like to express our sincere gratitude again for the invitation to submit a review paper for the Special Issue “Recent Advances in Thermal Energy Recovery and Utilization”. Enclosed is our revised manuscript with a revised title, “Actual Measurement and Evaluation of the Balance between Electricity Supply and Demand in Waste-treatment Facilities and Development of Adjustment Methods”.

Two experts from the field have reviewed our manuscript with very insightful feedbacks. We have addressed each comment made by the reviewers and incorporated it into our manuscript. We sincerely hope that the revised manuscript is now suitable for publication.

Correspondence about this paper can be directed to:

Prof. Hiroshi Onoda, Waseda University, Japan.

Thank you very much for your kind considerations.

Yours sincerely,

Daiki Yoshidome

Reviewer 2 Report

Recommendation

  1. Authors should use the following style of abstract: Background; Methods; Results; Conclusions.
  2. The Abstract Sections should not contain abbreviations. What does PV stand for?
  3. The Introduction Section: The current state of the research field should be carefully reviewed and key publications cited. Isn't 13 citations enough? In my opinion, the research field is not carefully reviewed.
  4. Line 28: [1] [2]. It does not comply with the instruction. The authors should write [1,2]. The same for all text.
  5. Authors should state the main aim of the work in the Introduction section.
  6. The Materials and Methods should be added and described with sufficient details.
  7. The Research Review Section is rather weak. It contains only four references [12-15].
  8. Table 5: “Power generation kW”. It must be corrected (put a comma in) and have the following form “Power generation, kW”.
  9. The Conclusion Section. The prospects for further research should be more specifically and clearly described.

Author Response

(The authors gave the same response as above.)

Round 2

Reviewer 1 Report

The caption of Figure 1 and that of Figure 2 are missing. Also, the results showed in Figure 1 need to be supported by suitable references

Author Response

Thank you for your comments. The part you pointed out has been corrected.

Reviewer 2 Report

Accept in present form

Author Response

Thank you for peer review. I fixed the missing references.